# Impact of Serum Lipid on Breast Cancer Recurrence

**DOI:** 10.3390/jcm9092846

**Published:** 2020-09-02

**Authors:** Sung Mi Jung, Danbee Kang, Eliseo Guallar, Jonghan Yu, Jeong Eon Lee, Seok Won Kim, Seok Jin Nam, Juhee Cho, Se Kyung Lee

**Affiliations:** 1Division of Breast Surgery, Department of Surgery, Samsung Medical Center, Sungkyunkwan University School of Medicine, Seoul 06351, Korea; happy12pm@hanmail.net (S.M.J.); jonghan.yu@samsung.com (J.Y.); jeongeon.lee@samsung.com (J.E.L.); seokwon1.kim@samsung.com (S.W.K.); seokjin.nam@samsung.com (S.J.N.); 2Center for Clinical Epidemiology, Samsung Medical Center, Sungkyunkwan University School of Medicine, 81 Irwon-ro, Gangnam-gu, Seoul 06351, Korea; cce.smc@samsung.com (D.K.); eliseo.guallar@samsung.com (E.G.); 3Department of Clinical Research Design and Evaluation, SAIHST, Sungkyunkwan University, Seoul 06351, Korea; 4Department of Health, Behavior and Society, Johns Hopkins Bloomberg School of Public Health, Baltimore, MD 21205, USA; 5Department of Clinical Research Design and Evaluation, SAIHST, Sungkyunkwan University, Seoul 06351, Korea

**Keywords:** Breast neoplasms, cholesterol, lipids, survival

## Abstract

The association between serum lipid level and prognosis of breast cancer is controversial. The purpose of this study was to evaluate the impact of serum lipid level in breast cancer recurrence. We analyzed a total of 4190 patients with operable breast cancer who had baseline serum lipid profiles; total cholesterol (TC), triglycerides (TG), low density lipoprotein-cholesterol (LDL-C), high density lipoprotein-cholesterol (HDL-C), apolipoprotein A-1, and apolipoprotein B. Recurrence-free survival is defined as the elapsed time from the date of curative surgery to the detection of any recurrence, and recurrence includes locoregional recurrence, distant metastasis, or both local and distant metastasis. Cox-proportional hazard analysis was used to estimate hazard ratios with 95% confidence intervals (CI) for study outcomes comparing the three lowest quartiles of each lipid parameter to the highest quartile adjusting for age, body mass index (BMI), and pathologic stage, estrogen receptor (ER), progesterone receptor (PR), comorbidities (hypertension, diabetes, or vascular event) at time of breast cancer diagnosis. Patients with dyslipidemia (high bad cholesterol and low good cholesterol level) had worse prognostic factors (i.e., negative hormone receptor status, positive human epidermal growth factor receptor 2 (HER2) expression, higher nuclear grade). After adjusting for these poor prognostic factors, the patients with dyslipidemia showed good prognosis for breast cancer recurrence. Our study showed that baseline high lipid level could be a good prognostic factor of breast cancer. This study indicates that desirable changes in lipid profile for cardiovascular disease risk are not always beneficial for patients with breast cancer. However, as proper control of lipid level has advantages for cardiovascular disease, these findings require careful interpretation.

## 1. Introduction

Breast cancer is not only the most common type of cancer in women [1], but also a representative cancer with a good prognosis. However, despite the fact that most breast cancers have a good prognosis, some still show bad progress such as recurrence and metastasis [2,3]. Breast cancer is a complex, heterogeneous disease with various biological characteristics, therefore, each breast cancer patient may have different responses to treatment [4]. Efforts have been made to find prognostic factors of breast cancer. Known major prognostic factors found so far include TNM stage, estrogen receptor (ER), progesterone receptor (PR), and human epidermal growth factor receptor 2 (HER2), however, most of these prognostic factors were for the character of the tumor itself. In addition, several studies have also examined the new prognostic factors related to individual characteristics, such as lifestyle (e.g., alcohol consumption, soy ingestion, active smoking, and obesity). Alcohol consumption increases the risk of breast cancer, and reducing alcohol consumption decreases the incidence of breast cancer among women who consume alcohol regularly [5]. Several studies have reported that soy consumption may protect against breast cancer, while other studies have shown that isoflavone, the major component of soy, enhance the proliferation of breast cancer cells [6,7]. Smoking was associated with increased breast cancer risk relative to all non-smokers [8], and being overweight is also related to a higher risk of mortality [9,10,11].

Among them, the role of blood cholesterol and statin on breast cancer prognosis was also suggested. Cholesterol is an essential structural component of the cell membranes which helps maintain both membrane structural integrity and fluidity. Cholesterol is also implicated in cellular signaling pathways through assisting in the formation of lipid rafts in the plasma membrane, which brings receptor proteins in close proximity with high concentrations of second messenger molecules [12]. These key cellular signaling is closely associated with malignant transformation of cells because of their role in organization of the cytoskeleton, cell migration, and angiogenesis [12,13,14]. In addition to this, within cells, cholesterol also serves as a precursor for the biosynthesis of steroid hormones, including the sex hormones progesterone, estrogens, and their derivatives which is important risk factors in breast cancer development. Because of the possibility of causal relations between cholesterol and breast cancer, the clinical role of cholesterol in cancer development and recurrence has been suggested. However, studies for the relation of cholesterol and breast cancer have not been concluded.

While dyslipidemia is common among breast cancer patients, the previous reports in the influence of dyslipidemia on breast cancer is inconsistent. Some studies found that the elevated levels of cholesterol were negative effects in breast cancer patients [15,16,17,18,19], otherwise, cholesterol level is not associated with the risk of breast cancer [20,21,22,23], and even showed an inverse correlation which is elevated serum levels of triglycerides (TG) may be associated with a reduced breast cancer risk [24]. In the field of lipoproteins which are responsible for the cholesterol transportation, studies are also inconclusive. Regarding high density lipoprotein-cholesterol (HDL-C), some authors showed a positive association between low HDL-C levels and increased breast cancer risk [21,25,26,27,28,29]; while others reported no association [30], and some even report a negative correlation between low HDL-C and breast cancer [31]. The study for low density lipoprotein-cholesterol (LDL-C) showed no association with breast cancer risk [21,23]. TG levels were no associated with risk of breast cancer [20,28], however, others showed an inverse correlation between high serum TG [32]. Therefore, this study aimed to evaluate the impact of serum lipid levels on breast cancer recurrence. 

## 2. Materials and Methods

### 2.1. Study Population

The study population was composed of patients who underwent breast cancer surgery at Samsung Medical Center Breast Cancer Center in Seoul, South Korea, from 1 January 2008 to 6 March 2014 (*n* = 8362). We excluded patients who were male (*n* = 40), those who received neoadjuvant therapy (*n* = 561), those with ductal carcinoma in situ (DCIS; *n* = 964) or stage 4 (*n* = 17) disease, and those with baseline dyslipidemia (*n* = 521), phyllodes tumor (*n* = 25), sarcoma (*n* = 4), metastatic cancer (*n* = 16), or malignancy of unknown origin (*n* = 4). Since the aim of the study was to evaluate the association between lipid profile at the time of diagnosis and five-year recurrence, we further excluded patients who did not have complete lipid profiles at the time of diagnosis (*n* = 2081) or with missing data for others (*n* = 2). Since study participants could have more than one exclusion criteria, the final sample size was 4190 and average follow up duration was 73.83 months (Figure 1). The Institutional Review Board of the Samsung Medical Center, Seoul Korea (IRB file no.2015-11-078) approved this study and waived the requirement for informed consent as we used only de-identified data routinely collected during routine cancer care. This article does not contain any studies animals performed by any of the authors. This study was performed in accordance with the Declaration of Helsinki.

### 2.2. Measurements

The baseline and clinicopathologic characteristics including demographic factors, pathologic findings, and perioperative treatment of breast cancer patients were collected from the electronic medical records. Age was collected at the time of breast cancer diagnosis and body mass index (BMI) was defined as the weight divided by the square of the body height at the time of breast cancer surgery. Menopause status was one of the following conditions; prior bilateral oophorectomy status, age over 60 years-old and amenorrhea for 12 or more months in patients under 60 years of age. Comorbidity for hypertension, diabetes, and vascular events were investigated.

Pathologic stage was based on American Joint Committee on Cancer 8th Edition [33]. Two experienced pathologists reviewed and determined characteristics of primary tumor based on size, axillary nodal status, resection margin and the status of receptors (ER, PR, and HER2) according to immunohistochemical (IHC) staining. Detailed information on surgery, adjuvant chemotherapy, radiotherapy, and hormone therapy were obtained from electronic medical records.

Participants were asked to fast for 12 h and avoid smoking on the morning of the examination. All lipids and lipoproteins were measured using a Hitachi 7600 Modular Dp-110 auto-analyzer (Hitachi, Tokyo, Japan) and included enzymatic colorimetric tests (total cholesterol (TC) and TG), homogeneous enzymatic colorimetric tests (LDL-C and HDL-C), and immunoturbidimetric assays (apolipoprotein A-1 (ApoA-1) and apolipoprotein B (ApoB)). 

In addition, we investigated recurrence through disease free survival, and recurrence-free survival is defined as the elapsed time from the date of curative surgery to the detection of any recurrence. Recurrence includes locoregional recurrence, distant metastasis, or both local and distant metastasis. Time to recurrence is defined as the interval between the date of surgical resection and the date of the first recurrence of the last follow-up. Overall survival is defined as the duration from the date of curative surgery until death by any causes. 

### 2.3. Statistical Analysis

Cox-proportional hazard analysis was used to estimate hazard ratios with 95% confidence intervals (CI) for study outcomes comparing the three lowest quartiles of each lipid parameter to the highest quartile. To consider potential confounding factors at baseline, we adjusted for age, BMI, pathologic stage, ER, PR and comorbidities (hypertension, diabetes, or vascular event) at time of breast cancer diagnosis. To evaluate predictive accuracies of risk prediction models, we used Harrell’s C-index which is the most widely used measure of predictive accuracy for survival models [34]. The values over 0.7 indicate a good model. All reported *p*-values were two-sided and the significance level was set at 0.05. All analysis was performed using SAS version 9.4 (SAS Institute, Inc., Cary, NC, USA) and STATA version 13 (StataCorp LP, College Station, TX, USA).

## 3. Results

### 3.1. Patients Characteristics

The mean age of study patients was 51.7 years and the mean BMI was 23.6 kg/m^2^. Overall, 72.2% of patients underwent breast-conserving surgery, and 75.3%, 69.1%, and 20.1% of patients were ER positive, PR positive, and HER2 positive, respectively. The percentage of patients on pathologic stage I, II, and III were 48.9%, 40.1%, and 11.0%, respectively. Lymphovascular invasion was present in 27.4% of patients, and 21.8% had multiple tumors. Regarding nuclear grade, 20.2%, 46.4%, and 33.3% of patients had low, intermediate, and high grade tumors, respectively (Table 1).

High levels of TC, LDL-C, and TG and low levels of HDL-C were clearly associated with older age, menopause, and high BMI. Higher levels of TC, LDL-C, and TG were statistically associated with negative expression of ER and PR, positive HER2, non-luminal subtype, and single lesion. Low HDL-C level was associated with negative ER expression alone. The tumors of patients with high LDL-C and low HDL-C showed a higher nuclear grade. Patients with high TG and low HDL-C level had higher disease stage, whereas TC and LDL-C level were not associated with stage (Appendix A).

There was no association between basal levels of serum lipid at the time of surgery and severity of the disease. When we calculated correlation between pathology stage which is severity of the disease and each blood marker, the correlation values were 0.01, −0.09, 0.04, 0.03, 0.04, and 0.05 in TC, LDL-C, HDL-C, TG, TG/HDL, and Non-HDL cholesterol, respectively. While some *p*-values were statistically significant, the magnitude is too small to tell the correlation (Figure 2).

### 3.2. Prognostic Role of Lipid Profile with Regard to Disease Recurrence in Breast Cancer Patients

The median follow-up period for recurrence was 22.28 (1.08–70.72) months. There were 125 recurrence events at the time of analysis (125/4190; 2.98%). As shown in Table 2, in the univariable analysis (crude hazard ratio in Table 2), the upper normal range (quartile III) of LDL-C level showed a 1.87-times (95% CI: 1.08 to 3.25) higher risk of recurrence compared to a high level of LDL-C (quartile IV). For TG level, the abnormally low group (quartile I) exhibited a 1.87-times (95% CI: 1.07 to 3.28) higher risk of recurrence compared to the abnormally high TG group (quartile IV). Conversely, a low level of HDL-C (quartile I) showed a protective effect on recurrence.

The C-index of the models for recurrence free survival at five years were 0.7568, 0.7573, 0.7489, 0.7613, 0.7545, and 0.7595 in TC, LDL-C, HDL-C, TG, TG/HDL and Non-HDL cholesterol, respectively. After adjustment for several confounding factors that had a high correlation with lipid profile and breast cancer prognosis, the prognostic effects of LDL-C, TG, and HDL-C on recurrence were maintained. In adjusted model, a high ratio of TG to HDL-C that could predict insulin resistance [35] and/or the development of coronary disease [36] showed a protective effect compared to a very low ratio.

Interestingly, as shown in Figure 3 and Figure 4, patients with a high level of bad lipids (LDL-C, TG, and TG/HDL) and low level of good lipid (HDL-C) generally displayed a good prognosis for breast cancer recurrence. There were patients with normal lipid level among patients with recurrence, and the patients with normal level of HDL (52.8%) or LDL cholesterol (70.4%) were relatively lower than patients with normal level of TC (90.4%) or TG (83.2%).

## 4. Discussion

In this study, breast cancer patients with a high level of bad lipids (LDL-C, TG, and TG/HDL) and low level of good lipid (HDL-C) showed good prognosis for cancer recurrence even though they had worse prognostic factors. 

Cholesterol is an essential structural component of the cell membranes which helps maintain both membrane structural integrity and fluidity. Cholesterol is also implicated in cellular signaling pathways through assisting in the formation of lipid rafts in the plasma membrane, which brings receptor proteins in close proximity with high concentrations of second messenger molecules [12]. These key cellular signaling is closely associated with malignant transformation of cells because of their role in organization of the cytoskeleton, cell migration, and angiogenesis [12,13,14]. Another evidence is 27-Hydroxycholesterol (27HC) which is a primary metabolite of cholesterol bind to an ER and Liver X receptor (LXR), and increases ER-dependent growth and LXR-dependent metastasis in mouse models of breast cancer [37]. In addition to this, within cells, cholesterol also serves as a precursor for the biosynthesis of steroid hormones, including the sex hormones progesterone, estrogens, and their derivatives which is important risk factors in breast cancer development.

However, the role of serum lipid in breast cancer development and prognosis is controversial. A correlation between serum lipid and breast cancer development was reported in several studies: some authors have shown that the levels of TC and TG were elevated in breast cancer patients [15,16,17,18], and others reported that a high level of LDL or very low density lipoprotein (VLDL) [29] and low level of HDL-C [21,25,26,31,32] are indicators of tumor development. Dyslipidemia is associated with overall breast cancer survival, and it has a role as prognostic factor for breast cancer [19]. However, there are also conflicting results. Some studies showed that no association between breast cancer risk and blood lipids was found [20,21,22,23], and others showed an inverse correlation between high serum lipid (TC [24], HDL [28,31], and TG [32]) and breast cancer development.

Although the role of serum lipids in the development of breast cancer has been relatively widely studied, there are few reports on their prognostic value. Laboratory investigations have shown that circulating levels of cholesterol, LDL-C, and HDL-C play a role in tumor development, growth, or aggressiveness and finally poor prognosis [19,38]. Bahl et al. [39] and Mousa et al. [40] showed a trend toward increased risk of recurrence with higher TC and LDL-C; however, another study reported that patients with hypercholesterolemia showed lower expression of endothelial vascular endothelial growth factor (VEGF) and basic fibroblast growth factor (bFGF) than patients with normocholesterolemia, and reducing blood cholesterol can suppress tumor growth by inhibiting tumor angiogenesis [41]. Studies on other types of cancer, not specifically breast cancer, have noted a relationship between low serum lipid and cancer morbidity and mortality [42,43]. 

Although there are still conflicting results about the serum lipid profile and breast cancer prognosis, in general breast cancer patients with high levels of TC, LDL-C, TG, and low level of HDL-C have been reported to show a poor prognosis of breast cancer, similar to the cardiovascular effects of lipid [19,38,39,40]. The inverse association that we observed between lipid profile and breast cancer recurrence appears to be discordant with previous reports. Our data indicated that desirable changes in lipid profile for cardiovascular disease risk are not always beneficial for breast cancer patients. Interestingly, patients with dyslipidemia (high bad cholesterol and low good cholesterol level) had worse prognostic factors of breast cancer (i.e., negative hormone receptor status, positive HER2 expression, higher nuclear grade). Even after adjusting for these poor prognostic factors, the patients showed good prognosis compared to patients with lower cholesterol levels. Among the variables in adjusted model, subtype (ER, PR, and HER2) is the factor which make change the significance. In breast cancer patients, IHC-defined subtypes showed different features, recurrence patterns, and survival [44].

There are several possible mechanisms underlying an inverse association between lipid profile and breast cancer recurrence. First, we could explain this result in terms of nutritional aspects. Serum lipid level is easily influenced by amount of food intake, and dietary pattern can also affect the lipid profile. A study that observed the effect of dietary patterns with respect to carbohydrate and fat content on blood lipid profiles in breast cancer patients showed that a low-fat diet resulted in a great reduction in TC and LDL-C, whereas a low-carbohydrate diet mainly influenced TG and HDL-C levels [45]. In addition, the serum lipid profile might be an indirect indicator of nutritional status [46,47]. Surprisingly, there are few studies on the role of poor nutritional status in prognosis of breast cancer. This is mainly because of the methodological difficulties in maintaining and applying lifestyle interventions to obtain direct evidence of a causal relationship in a randomized controlled trial. Although there is limited evidence suggesting that breast cancer prognosis is influenced by dietary fat intake and serum lipid level, indirect evidence is provided by observational studies with other indirect indicators such as BMI or lipid profile, which reflect the nutritional status. Obesity is a well-known risk factor and poor prognostic factor in breast cancer [9,10,11]. However, a large-scale study with Asian breast cancer patients reported that underweight was also associated with poorer survival [48,49]. The inclusion of undernourished patients in the lower lipid level group in our study, compared to the high lipid level group, may have partially contributed to the risk of recurrence.

Protective effects of serum lipid could also be explained by the indirect effects of statin usage. Although the data are controversial, there are several reports of the protective effects of statin usage in breast cancer patients [50,51,52]. We retrospectively searched past medical history and medication history and excluded patients who were diagnosed with dyslipidemia or using statins from our analysis. However, a complete enumeration survey was impossible because of the retrospective and single-institution study design. Many patients with dyslipidemia who were taking medicine in the local clinic rather than in our tertiary educational hospital could not be evaluated. In our study, 6% (521/8362) of patients were diagnosed with dyslipidemia. Compared to the generally accepted prevalence of approximately 20% [53,54], the prevalence of dyslipidemia in our study was extremely low. This implies that unrecognized dyslipidemia patients who were taking statins were probably included as normal patients in this analysis. In addition, even if all dyslipidemia patients were excluded in our analysis, the lipid levels analyzed in this study were baseline levels before surgery; therefore, the group of patients who showed good prognosis with dyslipidemia might have taken statins during the follow-up period. In such cases, the prognostic role of statin usage after breast cancer surgery will not be reflected in our results. This is a limitation of the cross-sectional study design.

There are other possible mechanisms to explain the good prognosis in patients with dyslipidemia. Ozdemir et al. [41] suggested that hypercholesterolemia impairs angiogenesis by suppressing endothelial and tumoral bFGF and VEGF expression. This could explain the good prognosis of hypercholesterolemia in our study. Other authors have suggested that hypocholesterolemia in cancer patients was caused by the breakdown of cholesterol by tumor cells expressing LDL receptor [55,56]. In this situation, the low level of serum lipid could be a reflection of tumor recurrence; therefore, these patients showed poor prognosis.

The strength of this study is its relatively large sample size. However, as already described, major limitations arise from the nature of the retrospective study design. For this reason, we could not obtain complete data on underlying disease such as diabetes and hypertension, which could influence the lipid profile, related diseases, and medication history. In addition, this was a cross-sectional study and therefore did not consider lipid profile changes during follow-up. We do not have any information regarding lipid marker at time of recurrence, therefore, our data did not account for the prognostic effects of lipid change and statin usage over time. Finally, our data were collected from one clinical institute and have limitations regarding generalization. It is necessary of further study regarding prognosis for the patients with recurrence considering importance and implication of patient with recurrence data.

## 5. Conclusions

Our study showed that baseline high lipid level could be a good prognostic factor of breast cancer. This indicated that desirable changes in lipid profile with respect to cardiovascular disease risk might not be beneficial for breast cancer patients. As proper control of lipid levels has advantages regarding cardiovascular disease, these data require careful interpretation.

## Figures and Tables

**Figure 1 jcm-09-02846-f001:**
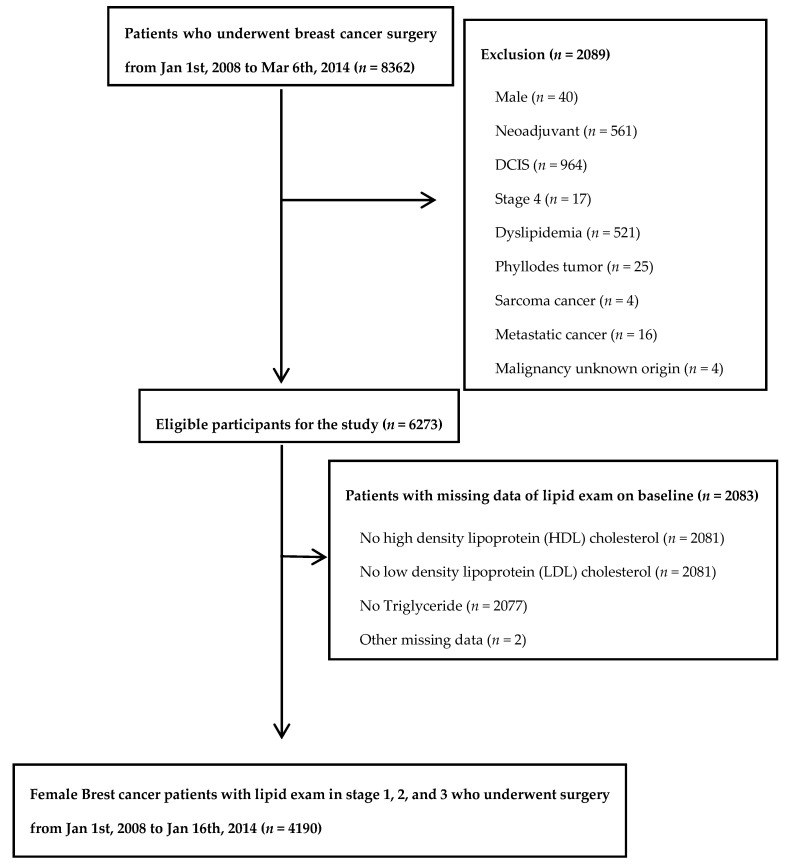
Flow chart of population.

**Figure 2 jcm-09-02846-f002:**
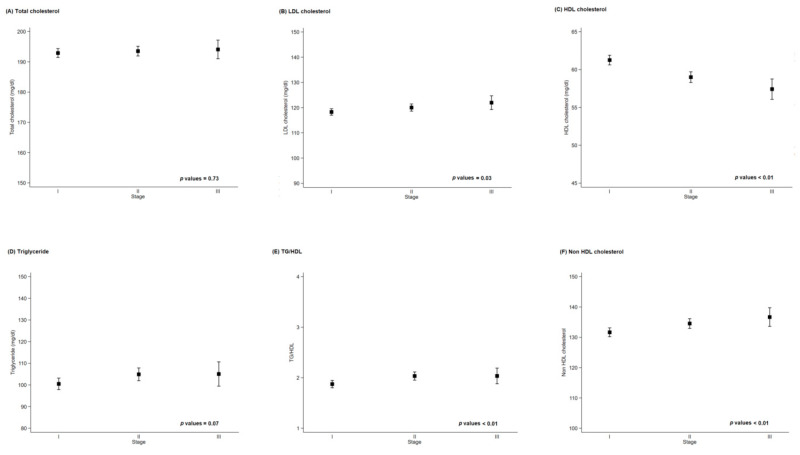
Baseline lipid marker by disease severity. HDL: high density lipoprotein; LDL: low density lipoprotein; TG: triglycerides.

**Figure 3 jcm-09-02846-f003:**
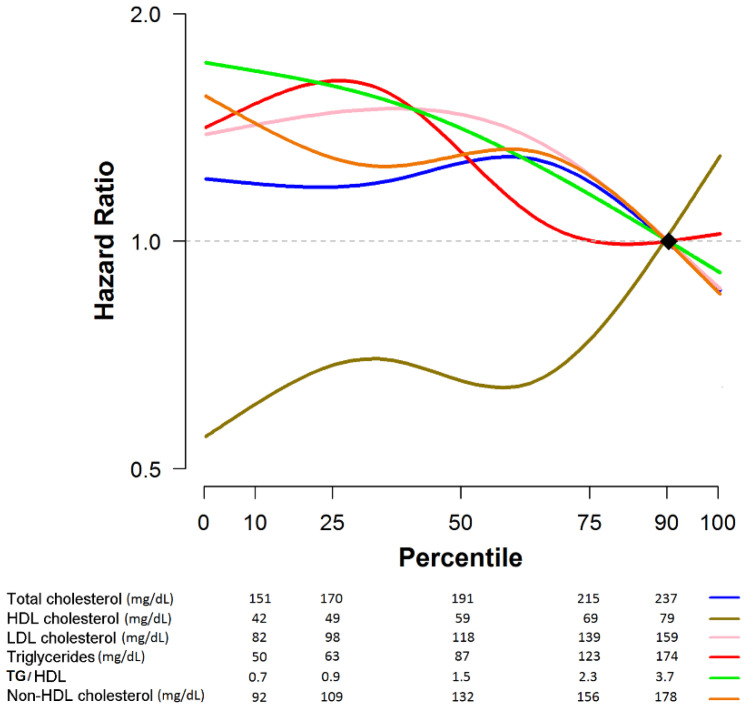
Hazard ratio by lipid exam score. HDL: high density lipoprotein; LDL: low density lipoprotein; TG: total cholesterol.

**Figure 4 jcm-09-02846-f004:**
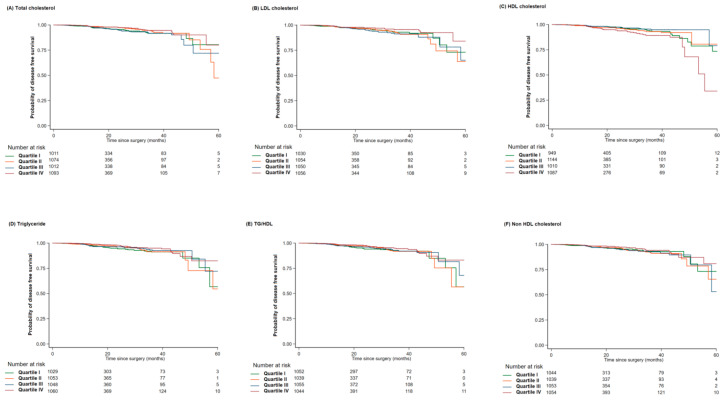
Hazard ratio by each lipid exam score. HDL: high density lipoprotein; LDL: low density lipoprotein; TG: total cholesterol.

**Table 1 jcm-09-02846-t001:** Patients characteristics.

	Overall (*N* = 4190)
	N (%)
**Age, mean (SD)**	51.7 (9.8)
**BMI, mean (SD)**	23.6 (3.2)
**Blood lab, mean (SD)**	
Total cholesterol (110–240) (mg/dL)	193.3 (33.7)
Triglyceride (50–200) (mg/dL)	102.8 (61.4)
HDL cholesterol (45–65) (mg/dL)	59.9 (14.8)
LDL cholesterol (40–130) (mg/dL)	119.4 (29.8)
TG/HDL	2.0 (1.7)
Non-HDL cholesterol	133.4 (33.7)
**Menopausal status**	
Yes	2217 (52.9)
No	1962 (46.8)
Unknown	11 (0.3)
**Comorbidity (*n* = 2946) ***	
No	2310 (78.4)
Yes	636 (21.6)
**Surgery type (*n* = 4188)**	
>Mastectomy	1166 (27.8)
BCS	3022 (72.2)
**Stage**	
I	2050 (48.9)
II	1679 (40.1)
III	461 (11.0)
**ER**	
Positive	3155 (75.3)
Negative	1032 (24.6)
Unknown	3 (0.1)
**PR**	
Positive	2894 (69.1)
Negative	1293 (30.9)
Unknown	3 (0.1)
**HER2**	
Positive	842 (20.1)
Negative	3275 (78.2)
Unknown	73 (1.7)
**Subtype**	
Luminal A	2737 (65.3)
Luminal B	406 (9.7)
HER2 type	436 (10.4)
TNBC	538 (12.8)
Unknown	73 (1.7)
**LVI**	
No	3018 (72.0)
Yes	1149 (27.4)
Unknown	23 (0.6)
**Multiplicity**	
No	3274 (78.1)
Yes	913 (21.8)
Unknown	3 (0.1)
**Nuclear grade**	
Low	847 (20.2)
Intermediate	1942 (46.4)
High	1393 (33.3)
Unknown	8 (0.2)
**Chemotherapy**	
No	1532 (36.6)
Yes	2586 (61.7)
Unknown	72 (1.7)
**Radiotherapy**	
No	882 (21.1)
Yes	3205 (76.5)
Unknown	103 (2.5)
**Hormone therapy**	
No	781 (18.6)
Yes	3409 (81.4)
AI ‡	1439 (34.3)
SERM §	1887 (45.0)

* Comorbidity included hypertension, diabetes, and vascular events and it was measured from 2009 year. ‡ AI (aromatase inhibitor) is Anastrozole, Exemestsane, Letrozole, and Arimidex. § SERM is Tamoxifen or Toremifen. BCS: breast-conserving surgery; BMI: body mass index; ER: estrogen receptor; HDL: high density lipoprotein; HER2: human epidermal growth factor receptor 2; LDL: low density lipoprotein; LVI: lymphovascular invasion; PR: progesterone receptor; SD: standard deviation; TG: triglyceride; TNBC: triple negative breast cancer.

**Table 2 jcm-09-02846-t002:** Prognostic role of lipid profile with regard to disease recurrence.

	Patients with Events	Crude		Adjusted	
		Hazard Ratio	*p*-Value	Hazard Ratio	*p*-Value
		(95% CI)		(95% CI)	
**Total cholesterol**					
Quartile I	34	1.49 (0.89, 2.48)	0.13	1.34 (0.78, 2.30)	0.28
Quartile II	29	1.18 (0.70, 2.01)	0.54	1.14 (0.66, 1.96)	0.63
Quartile III	36	1.60 (0.96, 2.65)	0.07	1.47 (0.88, 2.45)	0.14
Quartile IV	26	Reference		Reference	
**LDL cholesterol**					
Quartile I	33	1.68 (0.97, 2.90)	0.06	1.71 (0.96, 3.06)	0.07
Quartile II	35	1.72 (0.99, 2.95)	0.05	1.62 (0.92, 2.83)	0.09
Quartile III	36	1.88 (1.09, 3.22)	0.02	1.88 (1.09, 3.27)	0.02
Quartile IV	21	Reference		Reference	
**HDL cholesterol**					
Quartile I	33	0.58 (0.37, 0.93)	0.02	0.54 (0.33, 0.88)	0.01
Quartile II	33	0.65 (0.41, 1.04)	0.07	0.59 (0.37, 0.96)	0.03
Quartile III	20	0.43 (0.25, 0.75)	<0.001	0.45 (0.26, 0.77)	< 0.01
Quartile IV	39	Reference		Reference	
**Triglyceride**					
Quartile I	34	1.75 (0.05, 2.93)	0.03	1.88 (1.07, 3.29)	0.03
Quartile II	34	1.55 (0.93, 2.59)	0.09	1.66 (0.97, 2.82)	0.06
Quartile III	31	1.36 (0.81, 2.29)	0.25	1.32 (0.78, 2.25)	0.31
Quartile IV	26	Reference		Reference	
**TG/HDL**					
Quartile I	32	1.59 (0.95, 2.64)	0.08	1.82 (1.04, 3.21)	0.04
Quartile II	30	1.40 (0.83, 2.35)	0.21	1.57 (0.91, 2.73)	0.11
Quartile III	35	1.43 (0.87, 2.35)	0.16	1.45 (0.87, 2.42)	0.16
Quartile IV	28	Reference		Reference	
**Non-HDL cholesterol**					
Quartile I	31	1.41 (0.85, 2.35)	0.19	1.53 (0.89, 2.66)	0.13
Quartile II	32	1.37 (0.83, 2.28)	0.22	1.45 (0.85, 2.48)	0.17
Quartile III	34	1.47 (0.89, 2.43)	0.13	1.55 (0.92, 2.61)	0.10
Quartile IV	28	Reference		Reference	

Total cholesterol (mg/dL): $Q_{1}$ = 170, $Q_{2}$ = 191, $Q_{3}$ = 215; LDL cholesterol (mg/dL): $Q_{1}$ = 98, $Q_{2}$ = 118, $Q_{3}$ = 139; HDL cholesterol (mg/dl): $Q_{1}$ = 49, $Q_{2}$ = 59, $Q_{3}$ = 69; Triglyceride (mg/dl): $Q_{1}$ = 63, $Q_{2}$ = 87, $Q_{3}$ = 123; TG/HDL: $Q_{1}$ = 0.97, $Q_{2}$ = 1.47, $Q_{3}$ = 2.33 Non-HDL cholesterol: $Q_{1}$ = 109, $Q_{2}$ = 132, $Q_{3}$ = 156; Adjusted for adjusted for age, body mass index (BMI), pathologic stage, estrogen receptor (ER), progesterone receptor (PR), and comorbidities (hypertension, diabetes, or vascular event) at time of breast cancer diagnosis; CI: confidence interval; HDL: high density lipoprotein; LDL: low density lipoprotein; TG: triglycerides.

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
