# Peer review of "Impact of Serum Lipid on Breast Cancer Recurrence"

_jcm, 2020, doi:10.3390/jcm9092846_

Round 1

Reviewer 1 Report

Please provide basal level of lipids and its correlation with disease severity as figures for readers to understand (line 138-142). 

Author Response

Response to the Reviewer 1

  1. Please provide basal level of lipids and its correlation with disease severity as figures for readers to understand (line 138-142).

We included baseline lipid marker by disease severity as figure 4.

While some p-values were statistically significant, the magnitude is too small to tell the correlation.

Reviewer 2 Report

Revised manuscript address some of my previous queries regarding statistical analysis, method and discussion sections. I have no further queries for authors.

Author Response

Response to the Reviewer 2

Revised manuscript address some of my previous queries regarding statistical analysis, method and discussion sections. I have no further queries for authors.

Thank you.

Reviewer 3 Report

In this manuscript Jung, Kang and co-authors associated the lipid levels (TC, LDL, HDL , TG, TG/HDL and non HDL-C) at the time of BC surgery and recurrence.

I have few points in the Methods and Results section I would like to address, since many have been already covered.

Statistical Analysis:

I was wondering why you did not adjust for the menopause. Furtermore, how about the effect of the hormone therapy, the chemotherapy vs radiotherapy?

Did the glucose measurements were evaluated or used in anyway? Since you specified that you collected the data.  

The model 1 and 2, the adjustments is based on what? Sensitivity Analysis?

Why on Table 2 is the Quartile 4 as a reference? In the method section is written the Quartile I.

Why the albumin level is considered a marker of diet? In malnutrition might be very low, otherwise I would not use it as a main diet marker. Furthermore, its level that might be effected by several things, including medication (which could have been an advantage to know for the analysis). Why the levels of albumin are not reported in Table 1?

One last thing, when you proposed the two model adjustments is the model 2 which you trust the most? Do you know what is the main factor that makes changing the significance? 

Author Response

Response to the Reviewer 3

In this manuscript Jung, Kang and co-authors associated the lipid levels (TC, LDL, HDL , TG, TG/HDL and non HDL-C) at the time of BC surgery and recurrence.

I have few points in the Methods and Results section I would like to address, since many have been already covered.

Statistical Analysis:

  1. I was wondering why you did not adjust for the menopause. Furtermore, how about the effect of the hormone therapy, the chemotherapy vs radiotherapy?

Since the menopause status was highly correlated with age, and treatment were correlated with stage and hormone receptor, we did not include these factors in the model.

  1. Did the glucose measurements were evaluated or used in anyway? Since you specified that you collected the data.

Glucose measurements were not included as variables. We revised the methods section.

  1. The model 1 and 2, the adjustments is based on what? Sensitivity Analysis?

Model 1 were included confounding factors among basic characteristics of study participants, and model 2 were included additional factors which we consider as confounding factors. With respect to the reviewer’s comment, now we had deleted model 1 and included unadjusted and adjusted models.

  1. Why on Table 2 is the Quartile 4 as a reference? In the method section is written the Quartile I.

Since it was a typo, we revised the methods section.

  1. Why the albumin level is considered a marker of diet? In malnutrition might be very low, otherwise I would not use it as a main diet marker. Furthermore, its level that might be effected by several things, including medication (which could have been an advantage to know for the analysis). Why the levels of albumin are not reported in Table 1?

As reviewer’ s comments, we deleted the albumin in the model and the results were similar.

With respect to the reviewer’ comment, now we updated relevant tables (Table 2) and descriptions in the Methods sections.

Methods (page 8, lines 148-151)

“To consider potential confounding factors at baseline, we adjusted for age, BMI, pathologic stage, estrogen receptor (ER), progesterone receptor (PR), and comorbidities (hypertension, diabetes, or vascular event) at time of breast cancer diagnosis.”

Table 2. Prognostic role of lipid profile with regard to disease recurrence.

Patients with Events

Crude

Adjusted

Hazard Ratio

p-Value

Hazard Ratio

p-Value

(95% CI)

(95% CI)

Total cholesterol

Quartile I

34

1.49 (0.89, 2.48)

0.13

1.34 (0.78, 2.30)

0.28

Quartile II

29

1.18 (0.70, 2.01)

0.54

1.14 (0.66, 1.96)

0.63

Quartile III

36

1.60 (0.96, 2.65)

0.07

1.47 (0.88, 2.45)

0.14

Quartile IV

26

Reference

Reference

LDL cholesterol

Quartile I

33

1.68 (0.97, 2.90)

0.06

1.71 (0.96, 3.06)

0.07

Quartile II

35

1.72 (0.99, 2.95)

0.05

1.62 (0.92, 2.83)

0.09

Quartile III

36

1.88 (1.09, 3.22)

0.02

1.88 (1.09, 3.27)

0.02

Quartile IV

21

Reference

Reference

HDL cholesterol

Quartile I

33

0.58 (0.37, 0.93)

0.02

0.54 (0.33, 0.88)

0.01

Quartile II

33

0.65 (0.41, 1.04)

0.07

0.59 (0.37, 0.96)

0.03

Quartile III

20

0.43 (0.25, 0.75)

<0.001

0.45 (0.26, 0.77)

< 0.01

Quartile IV

39

Reference

Reference

Triglyceride

Quartile I

34

1.75 (0.05, 2.93)

0.03

1.88 (1.07, 3.29)

0.03

Quartile II

34

1.55 (0.93, 2.59)

0.09

1.66 (0.97, 2.82)

0.06

Quartile III

31

1.36 (0.81, 2.29)

0.25

1.32 (0.78, 2.25)

0.31

Quartile IV

26

Reference

Reference

TG/HDL

Quartile I

32

1.59 (0.95, 2.64)

0.08

1.82 (1.04, 3.21)

0.04

Quartile II

30

1.40 (0.83, 2.35)

0.21

1.57 (0.91, 2.73)

0.11

Quartile III

35

1.43 (0.87, 2.35)

0.16

1.45 (0.87, 2.42)

0.16

Quartile IV

28

Reference

Reference

Non HDL cholesterol

Quartile I

31

1.41 (0.85, 2.35)

0.19

1.53 (0.89, 2.66)

0.13

Quartile II

32

1.37 (0.83, 2.28)

0.22

1.45 (0.85, 2.48)

0.17

Quartile III

34

1.47 (0.89, 2.43)

0.13

1.55 (0.92, 2.61)

0.10

Quartile IV

28

Reference

Reference

Total cholesterol (mg/dl): =170, =191, =215; LDL cholesterol (mg/dl): =98, =118, =139; HDL cholesterol (mg/dl):=49, =59, =69; Triglyceride (mg/dl): =63, =87, =123; TG/HDL: =0.97, =1.47, =2.33 Non HDL cholesterol: =109, =132, =156; Adjusted for adjusted for age, BMI, pathologic stage, estrogen receptor (ER), progesterone receptor (PR), and comorbidities (hypertension, diabetes, or vascular event) at time of breast cancer diagnosis

While the baseline levels of albumin could find in Table 1 (Albumin (g/dl), mean (SD) = 4.5 (0.3)), we also deleted this in the Table 1.

  1. One last thing, when you proposed the two model adjustments is the model 2 which you trust the most? Do you know what is the main factor that makes changing the significance?

As reviewer’s comments, model 2 is the final model in this study. Among the variables, subtype (ER, PR and HER2) is the factor which make change the significance. With respect to the reviewer’s comment, now we only include model 2. Please also see the response to the comment #3.

Round 2

Reviewer 3 Report

The revised manuscript addresses all my points, therefore I have no further queries for the authors.

This manuscript is a resubmission of an earlier submission. The following is a list of the peer review reports and author responses from that submission.

Round 1

Reviewer 1 Report

In this manuscript, authors evaluated the role of serum lipids (cholesterol, LDL, HDL, TG) on the progression and recurrence of breast cancer. The study has its own advantage and have the potential to help clinician to manage the recurrence of the breast cancer. However, there are serious concerns with respect to study designs and the way data presented which have led down the impact of the study and final conclusion. Some are mentioned below which can be used to revise the manuscript.

For example, in this study there are two components, progression of the breast cancer and recurrence of the breast cancer. Authors failed to describe the effect of serum lipids separately on each aspect.

With respect to progression of the breast cancer authors must have described how the basal levels of serum lipids at the time of surgery correlated with the severity of the disease. And it should be compared with control population which is completely missing in this study.

Also, with respect to recurrence, a comparison of the basal serum lipids levels with serum lipids levels at the time recurrence must be made and presented clearly. In the same line, it is not clearly mentioned in the manuscript that blood was collected at two time from the same individual once at the time surgery and second at the time of recurrence.

In this manuscript results are presented in tables which is good but difficult to understand for research community. Authors must include multiple figure to explain the association of serum lipids with both progression and recurrence of disease.

The effect of diet, life style and multiple disorders for example diabetes are not considered for recurrence of the disease. 

Reviewer 2 Report

The paper titled "Impact of serum lipid on breast cancer recurrence" describes the cross sectional study analysis to evaluate the role of lipid profile as a prognosis marker in breast cancer, more specifically recurrence of cancer.
Overall, I find the research interesting and can have potential contributory value towards global search of identifying accurate clinical biomarkers for predicting breast cancer progression and recurrence. However, I think the research could have been better presented and methods and conclusions clearly described.
I think manuscript can be further improved in following areas before consideration for publication,
1) Abstract-- need language and grammar changes with clearly outlining the findings and their potential implications of the findings in breast cancer research/clinical practice.
2) Introduction: Introduction need few language and grammatical improvements.
3) Methods-- Needs clear descriptions, in statistical part, regarding two models utilized in the study how they are adjusted for different variables.
4) There are redundancies in sentences between introduction and discussion section.
Few comments and questions for authors regarding result section:
1) What are the specificity and accuracy of the models used in the study ?
2) As there are multiple analysis run on datasets what would be the FDR corrected P values for those analysis?
3) Where the patients (among those 125) with normal lipid profile would fall? or in other words are there any patients with normal lipid levels but still showed recurrence of tumor ?
4) Did authors also looked at the prognosis for the rest of patients (all -125) after training the models using findings of their study from 125 patients data ? would be interesting to see what would be the predictive accuracy of the models!
5) Additional discussion can be included regarding potential utility or implications of this work for future breast cancer research or for clinical practice.